# The Role of Molecular Imaging in Personalized Medicine

**DOI:** 10.3390/jpm13020369

**Published:** 2023-02-19

**Authors:** Suliman Salih, Aisyah Elliyanti, Ajnas Alkatheeri, Fatima AlYafei, Bashayer Almarri, Hasina Khan

**Affiliations:** 1Radiology and Medical Imaging Department, Fatima College of Health Sciences, Abu Dhabi 3798, United Arab Emirates; 2National Cancer Institute, University of Gezira, Wad Madani 2667, Sudan; 3Nuclear Medicine Division of Radiology Department, Faculty of Medicine, Universitas Andalas, Padang 25163, Indonesia; 4Radiology Department, Sheikh Shakhbout Medical City, Abu Dhabi 3798, United Arab Emirates

**Keywords:** precision medicine, molecular imaging, radionuclides, nuclear imaging

## Abstract

The concept of personalized medicine refers to the tailoring of medical treatment to each patient’s unique characteristics. Scientific advancements have led to a better understanding of how a person’s unique molecular and genetic profile makes them susceptible to certain diseases. It provides individualized medical treatments that will be safe and effective for each patient. Molecular imaging modalities play an essential role in this aspect. They are used widely in screening, detection and diagnosis, treatment, assessing disease heterogeneity and progression planning, molecular characteristics, and long-term follow-up. In contrast to conventional imaging approaches, molecular imaging techniques approach images as the knowledge that can be processed, allowing for the collection of relevant knowledge in addition to the evaluation of enormous patient groups. This review presents the fundamental role of molecular imaging modalities in personalized medicine.

## 1. Introduction

Hippocrates of Kos (c. 460—ca. 370 BC) stated that it is more important to understand the type of person with a disease than to identify the type of disease the patient has. This statement by the father of modern medicine is considered the platform of personalized medicine (PM). The term “personalized medicine” refers to a relatively new field of medicine that aims to enhance diagnostic precision and reduce therapeutic failures. There is wide use of molecular imaging modalities in screening, diagnosis, treatment, assessment of disease heterogeneity, progression planning, molecular characteristics, and long-term follow-up for various diseases. As opposed to conventional imaging techniques, molecular imaging approaches images as data that can be mined and used to extract additional information as well as assess large populations of patients [1,2].

Molecular imaging has become widely used in many diseases, with a particular focus on cancer care. It refers to the in vivo characterization and measurement of key biomolecules and molecular events underlying malignant conditions. As applied to oncology, this article discusses both established and emerging methods of molecular imaging. Current molecular imaging techniques offer advantages for improving clinical cancer care as well as drug development [1,2,3].

Cancer is one of the challenging issues in public health care that affects millions of people around the world and leads to high mortality rates worldwide. According to the World Health Organization, since cancer is positively correlated with patients’ ages, even if it affects younger people, cancer incidence prevalence and mortality are expected to significantly raise due to population growth and aging. This increases the importance of promoting and advancing health care systems and strategies and developing conventional therapy methods, including chemotherapy, radiation therapy, and surgery, to eliminate cancer cells, increase the survival rates of patients, and ensure sustainability, providing satisfactory global healthcare based on prevention, accurate diagnosis, and effective treatment that has less multi-drug resistance, high selectivity, and less cytotoxicity [4,5,6].

Normal healthy cells do not evolve to be cancerous but are established from an aggregation of DNA damage to cells. Cancer cells have a complex pathophysiology and determining the main cause for each case is difficult to achieve; therefore, there are three main categories that are used to determine cancer causes. First, genetic inheritance and epigenetic factors; second, physical agents, such as exposure to genotoxic chemicals, UV light, and ionizing radiation; third, physiology, namely metabolic changes and telomerase enzyme activity changes [4].

Annually, there are 24 million new patients who are diagnosed with cancer worldwide; usually, there are multiple causes, manifesting differently over time, varying from one patient to another, which makes treating cancer very challenging. Conventional cancer therapy is non-targeting and damages healthy cells due to the accumulation of therapy on them, leading to the reduced efficiency of treatment. Therefore, it is desirable to develop targeted drug delivery systems that can deliver the therapeutic agent to the target tumor to reduce the adverse effects and improve efficacy [4,5,6].

Significant rapid advances in molecular biology, cancer biomarkers, and radio-genomics help to have a better understanding of cancer, resulting in developing personalized medicine and molecular imaging since both are strongly dependent on the collaboration of different clinical disciplines. Personalized medicine is a comparatively new emerging practice of medicine that focuses on providing the tumor genetic profile to proffer individual prevention, diagnosis, and treatment, which reflects on cancer treatment by improving the anti-cancer therapeutic efficiency and reducing the adverse effects. Molecular imaging is used widely in screening, detection and diagnosis, treatment, assessing disease heterogeneity and progression planning, molecular characteristics, and long-term follow-up. Moreover, it is able to detect very tiny tumors and assess their activity numerically, which makes molecular imaging one of the most scientific reasons that contributes greatly to expanding and developing the personalized medicine, research, clinical trials, and medical practice of cancer fields, evolving a new generation of platforms with greater accuracy and sensitivity for in vivo quantification and characterization of various biological processes [4,6,7,8,9,10,11,12].

This paper will review a wide range of published research on personalized medicine and molecular imaging to define the role of molecular imaging (ultrasound, MRI, PET-CT, PET-MRI, SPECT) in personalized medicine. In addition, the article discusses the importance of molecular imaging to the emerging field of theranostics and how molecular imaging may one day be integrated with other diagnostic techniques to improve cancer treatment efficiency and effectiveness [1,2,3].

## 2. Ultrasound (US)

Nowadays, personalized medicine involves non-invasive imaging methods to detect pathologies and treat patients. These imaging methods can be divided into two groups: morphological/anatomical imaging and molecular imaging based on nanobodies. Multidisciplinary collaboration across several domains, including radiology, nuclear medicine, pharmacology, chemistry, molecular and cell biology, physics, mathematics, and engineering, has led to the creation and translation of molecular imaging [5,13,14,15,16,17].

Ultrasound is a high-resolution structural imaging technique that is one of the most widely used diagnostic clinical imaging modalities. Ultrasound is an imaging technique and biological system approach that can be used as two highly efficient methods for thermal cancer therapy therapies (thermoablation and high-intensity frequency ultrasound treatment (HIFU) that produce hyperthermia or hypothermia). It can also be used for diagnosis in clinical trials due to its beneficial properties such as its safety due to no ionizing irradiation, wide availability, portability, real-time imaging/acquisition time (min), high spatial resolution, external or internal application (endoscopy), inexpensiveness, high sensitivity, and ability to be combined with contrast agents to separate contrast and morphological imaging (with use of harmonics). These characteristics improve interest in wide ultrasound application and its role in personalized medicine [8,18,19,20,21].

An ultrasound method called speckle tracking echocardiography (STE) is employed to evaluate myocardial function. This technique examines how distinctive speckle patterns, natural myocardial acoustic markers, move during the cardiac cycle. Myocardial velocities and intrinsic cardiac deformation can be calculated offline (strain and strain rate). This strategy should be used in patient follow-up even though it necessitates specialized software and depends on better image quality [22].

Another method called ultrasound elastography (USE) is a new tool for measuring tissue stiffness and has demonstrated the importance of tissue elasticity for the diagnosis of tumors. It measures the spatial variation of the mechanical response and monitors local changes in tissue pressure during anti-stromal therapy and hyaluronic acid depletion [23,24]. In the clinical environment, it is used as a non-invasive assessment of liver fibrosis to characterize breast masses, evaluate the thyroid nodules, target biopsy-facilitated prostate, characterize focal renal lesions, and kidney and lymph node imaging are emerging [25,26,27,28]. Nanoplatforms have been used in every biomedical imaging modality, including ultrasound molecular imaging with microbubble agents, formerly known as blood pool contrast. In ultrasound, they have been used to provide early detection, accurate diagnosis, monitor the delivery and uptake of therapeutic agents in real-time, and facilitate individualized therapy of diseases using the contact-facilitated drug delivery mechanism, which is based on highly lipophilic agents such as paclitaxel contained within emulsions and relies on close apposition between the agents. The other mechanism is the liposomal drug delivery mechanism that involves lipid exchange or lipid mixing between the emulsion vesicle and the targeted cell membrane, which depends on the extent and frequency of contact between two lipidic surfaces. Some of the ultrasound contrast is summarized in (Table 1). The US employs high-frequency sound waves emitted from a transducer placed against the skin and reflected differently by different organs and tissues. The contrast of ultrasound is dependent on the sound speed, sound attenuation, backscatter, and imaging algorithm [19,29,30,31].

This imaging technique has played a role in therapeutics and theragnostic by determining the presence and degree of molecular targets for a certain disease and confirming the effective administration of these drugs; it is used for the targeted delivery of drugs, including genetic material, which is proliferating. It is also used for focal disruption of the blood–brain barrier to enable access to the brain for hydrophilic diagnostic and therapeutic agents, in blood pool enhancement, perfusion imaging, characterization of lesions, phase and metabolism, echocardiography, monitoring and quantifying arthritis, identifying the phase of this heterogeneous disease, and detecting sites of atherosclerosis pathogenesis before lesions occur by accumulating the microbubbles in ECAM-rich sites [12,30,45,46,47].

Otherwise, with all the incredible use of ultrasound contrast agents but still limited to the vasculature, they produce a high background signal and “tether” to a surface, which limits their ability to oscillate, and thus their echogenicity is slightly hampered. The FDA reported microbubble-based injectable suspensions (e.g., Perflutren Lipid Microsphere) and Optison (Perflutren Protein-Type A Microspheres for Injection) in 2007 due to the risk of serious cardiopulmonary reactions during or within the first 30 min after administration. However, these warnings were edited in 2008, but these microbubble contraindications limited their clinical use. Moreover, the US has poor penetration specificity and limited sensitivity, meaning it detects tumors without being able to distinguish between malignant and non-malignant tumors. It also has poor image quality since the blood is a poor scatterer at clinical diagnostic transmission frequencies. Currently, it is not possible to use US to scan the full human body, and it is considered operator dependent. All the listed limitations can weaken US applications in personalized medicine [6,15,48,49,50].

Contrarily, a lot of work is being put towards expanding US’s function in molecular imaging and personalized medicine. Some potential uses are the incorporation of sound pulses into MRI pulse sequences to broaden US’s use of the molecular field, drug delivery by increasing the rate of lipid exchange and the tendency for fusion or improved contact between the nanoparticles and the targeted cell membrane while using the safe level of US waves, drug-loaded albumin-based carriers, and controlling drug release. Additionally, researchers have improved contrast-enhanced ultrasounds to detect image molecular markers and to boost ultrasound image quality, making them appropriate for ultrasound therapy [8,15,51,52,53,54,55].

## 3. Magnetic Resonance Imaging (MRI)

Molecular imaging is a turning point in the era of personalized imaging. The term was introduced in the last decade, referring to a variety of scientific disciplines, from biomedicine to clinical medicine, and it explains numerous clinical issues. The development of multimodality imaging, such as multiparametric magnetic resonance imaging (MRI) and PET/MRI, which are used more frequently for cancer diagnosis, staging, and surveillance, is one of the main components of molecular imaging. This type of imaging is becoming more important since it supports how physicians formulate diagnoses and arrange treatments in cases where ineffective surgical operations and harmful treatments might be avoided [15,19,48,56,57].

Magnetic resonance imaging (MRI) is based on the interaction of specific nuclei, usually protons, with molecules neighboring each other in intercellular tissue. The relaxation times of various tissues vary, which causes an endogenous contrast. External contrast agents can further improve this by specifically reducing the length of either the longitudinal T1 relaxation or the transverse T2 relaxation [12,18,52].

All types of testing for analyzing tumor behavior have been demonstrated to be affected by molecular imaging techniques. Signaling mechanisms that support cancer cell reprogramming are essential because the clonal heterogeneity of the tumor determines the sort of treatment required for each individual cell, thus “personalizing” the course of treatment. Fluorodeoxyglucose, the glucose analogue on a combined PET-MRI modality, is a significant molecular marker used in the in vivo visualization, characterization, and quantification of biological processes in a tumor at the molecular and cellular level. In cancer, when information on the morphology and function of a diseased lesion is co-registered, the innovation involved in merging these modalities enhances its accuracy [15,21,45,47,58,59].

Functional molecular imaging offers a distinctive perspective on various conditions. We are able to examine both the underlying biochemistry and the geographical and temporal changes in biomarkers using techniques such as magnetic resonance imaging. When imaging and molecular diagnostics are used together, it becomes possible to measure abnormal cellular signalling pathways in unprecedented depth. This succinct essay demonstrates how radiotracers and nuclear imaging techniques are being developed to track drug effectiveness and, at the same time, promote the objective of individualized healthcare [47,49,59,60,61].

## 4. Single-Photon Emission Computerized Tomography (SPECT)

SPECT is one of the molecular imaging techniques that has really helped to visualize, characterize, and measure abnormal biologic processes of cancer at the macro and micro levels by emitting gamma rays at various energies to detect the radiopharmaceuticals (Table 2) that provide sensitivity, the ability to quantify their uptake, and detection of these agents at any depth in the body, which makes it appropriate for the monitoring of progress and outcome after surgery, radiotherapy, or chemotherapy. The creation of tailored theragnostic agents using the non-invasive imaging technique known as SPECT would make it possible to choose patients more precisely and improve the discovery, delivery, and development of new drugs. Despite these benefits, SPECT has a low resolution, poor contrast, and lacks markers for anatomical and physiological differences in biodistribution because correctly localizing anatomical references for lesions is difficult [12,17,21,47,62,63,64,65,66].

### 4.1. Hybrid SPECT

By boosting sensitivity with molecular targeting, anatomic specificity, and resolution, hybrid technologies such as SPECT/CT, PET/SPECT, and SPECT/MRI enable further advancement in molecular imaging. PET/SPECT is an infusion technology that uses ligands with short half-lives to provide long-term cell vision for up to 14 days. It has the potential to identify molecular targets that are significant in the progression of the disease and theragnostic strategies. Nanomolecular diagnosis of the severity and distribution of the disorders is provided by SPECT/MRI. SPECT/MRI has received less focus in pre-clinical research than PET/MRI [17,63,67,69,71].

### 4.2. SPECT/CT

The benefits of SPECT/CT include a quick acquisition time of 1–2 min, low radiation exposure to the patient, and a good signal-to-noise ratio that allows for the use of the scout image for reasons other than attenuation correction. These factors provide the necessary pretherapy information on biodistribution, dosimetry, the limiting of critical organs or tissues, and the maximum tolerated dose, making the tailored therapy and imaging of personalized medicine safe and appropriate. The main strength of SPECT/CT is the features for individual modalities that increase the molecular and anatomical integration systems. SPECT/CT have a significant impact on clinical management by better guiding subsequent operations, avoiding unnecessary procedures, and providing predictive information that aids in guiding change in both intra- and inter-modality therapy [2,17,52,64,67,70].

## 5. Positron Emission Tomography (PET)

PET is the gold standard in clinical molecular imaging because it possesses the high sensitivity required for deep tissue penetration and visualization of most interactions between physiological targets and ligands. Due to this, non-invasive detection up to the picomolar level is achievable. By producing quantitative images and 3D morphological images at quick scan times, which enables dynamic imaging (time-resolved images to be generated), it has become the fastest-growing clinical imaging technology and is now a current tool in cancer diagnoses and cancer treatment planning. The basis of the PET technique is the phenomenon of positron–electron annihilation, resulting in the formation of two high-energy photons (511 keV) emitted in opposite directions (180°). PET using biomarkers are labelled with positron (a positively charged electron)-emitting radioisotopes, primarily nitrogen, oxygen, carbon, and fluorine, which are short-lived elements (2–110 min) used to image the molecular interaction of biological processes such as cell proliferation, glucose metabolism, amino acid uptake, and membrane biosynthesis. They also deliver information about biomarker expression and tissue biochemical characteristics, provide the exact location of a lesion frequently before symptoms arise, determine molecular phenotypes, provide valuable molecular, functional, and metabolic information, and aid in determining the tumor biology of neoplasms by creating quantitative imaging that is capable of transforming collected gamma rays into quantitative terms. These quantitative images support safer surgical resections that minimize morbidity and mortality as well as increase the cost-effectiveness of healthcare with a measurable return on investment. They also aid in the diagnosis, optimization, and personalization of treatment for a variety of diseases. Moreover, the use of several tracers in PET technology is one of the technique’s distinctive advantages (Table 3). Over the past decade, the clinical use of PET has increased dramatically. The most often used glucose analog is 18F-fluorodeoxyglucose (FDG). Some novel receptor-active peptides have found usage in the transport and phosphorylation of FDG, but then the FDG is stuck [2,4,10,12,14,17,20,21,23,47,49,53,56,60,72,73,74,75,76,77,78,79].

Due to the special characteristics of PET and the quick development and growth of hybrid PET in recent decades, the scope of PET clinical applications has increased. By advancing the clinical use of PM, PET clinical applications will continue to support the role of molecular imaging in the era of personalized medicine. PET has been used in oncology using antimetabolic image information for diagnosis and identifying undetected distant metastases, stages, and volume in cancers and the presence of inflammatory infiltrate. By providing personalized medicine, such as personalized chemotherapy, immunotherapy, targeted therapy, and dosage, as well as personalized evaluation of response early in treatment due to changes in glucose metabolism and evaluation of the antiangiogenic therapeutic result, PET scanning can improve cancer management. PET scans have many advantages in toxicology studies because they are an important tool in personalized drug discovery and development, screening, identifying new drug candidates, and evaluating individual patient susceptibility to treatment by nanocarrier systems. PET imaging is ideal for radiopharmaceutical micro-dosing research and drug therapy development. Additionally, it contributes to minimizing expenditure on medication development and animal use in preclinical toxicological research. In order to determine whether the drug concentration delivered to the target is sufficient to elicit a pharmacologic response, the PET imaging protocol can be used to measure both AR levels (in the sense of a predictive biomarker for estimating response to therapy and monitoring drug-target engagement) and AR activity using a PD biomarker [1,12,21,52,58,72,73,75,78,79,87,89,90,91,92].

### 5.1. PET-US

Hybrid imaging adds value to imaging data and provides efficient diagnosis, radiogenomics, and therapy planning. PET combined with various modalities such as US, MRI, optical imaging systems, immune probes, and CT, are most commonly used in clinical settings today. PET-US uses radiolabelled microbubble shells such as 18F-labeled, albumin-shelled, and VEGFR2-targeted, which have a short half-life and are several micrometres in size. This modality can be used for investigating the biodistribution of microbubbles after i.v. injection and offers better quantification, which is particularly true in biodistribution analyses and can be used for targeted drug delivery, such as delivering VEGFR2 in breast cancer [2,31,54,80].

### 5.2. PET-MRI

PET-MR units are currently in development and being used in pre-clinical environments. This dual modality allows high spatial resolution, temporal resolution and accuracy, superior soft tissue contrast and multi-planar capabilities, and less ionizing radiation exposure. These features allow it to perform translational research from a cell culture setting to pre-clinical animal models to clinical applications, which is advantageous for the drug discovery and evaluation process that could help optimize the development of new drugs non-invasively and develop radiotracers. Additionally, it has been used to measure processes as diverse as blood flow and volume, tissue oxygenation, tissue pH, protein synthesis, cellular proliferation, enzyme kinetics, endogenous metabolite concentration, water diffusion, tissue anisotropy, vascular permeability, and better treatment response, providing information on downstream effects from multiple pathways, even though it is more limited with respect to the number of molecular processes that can be imaged, and provides additional opportunities for facilitating targeted biopsy and the determination of its efficacy. Hybrid PET/MR systems provide complementary multi-modal information about perfusion, metabolism, receptor status, and function, together with excellent high-contrast soft tissue visualization without the need to expose the patient to additional radiation, which makes them very useful for precision medicine cancer care in cardiac sarcoidosis, degenerative diseases such as Alzheimer’s disease, and cancers such as pharyngeal and ovarian cancer [2,12,14,19,21,31,48,50,58,75,76,77,80,93,94].

### 5.3. Positron Emission Tomography-Optical Imaging (PET-OI)

PET and optical imaging have been combined and demonstrated in vitro, ex vivo, or in vivo in recent years. The principal benefits are related to the combination of increased tissue penetration of radiation from positron emitter radionuclides that enables non-invasive quantitative imaging and tumor detection and light generated by the fluorescent probe for optical imaging during surgery, in particular, robotic surgery. This allows for effective, targeted drug delivery in vivo without causing systemic toxicity, and both the administered dose and therapeutic efficacy can be precisely monitored non-invasively over time. In order to image and evaluate the concentration and function of the target without having an impact on it, the probe is utilized in extremely low mass amounts during PET imaging (tissue concentrations of around femtolitres per gram of tissue). Similarly to PET/MRI, this dual imaging is used in the drug development process to identify, accurately measure, and assess medications’ performance in vivo in mice models and human patients. This will make it possible to discover drugs more successfully using a systems-based approach that is driven by molecular imaging and diagnostic techniques [2,59,94].

### 5.4. Immuno-PET

Despite not being fully realized, the combination of radiation therapy and immunotherapy has the potential to change the field of oncology. Immuno-PET imaging could play a critical role in providing the crucial information required to help understand this sophisticated connection. Nowadays, immuno-PET is a safe multimodality treatment strategy that helps to move toward precision medicine using radio-labelled antibodies and targets that combine with the high sensitivity and quantitative potential of PET non-invasively to provide quantitative, high quality, high spatial, and temporal resolution images that help to estimate the antigenic expression level of immuno-PET such as immune checkpoints and effector molecules, or the detection and tracking of immune cell populations such as T-cell subsets and chimeric antigen receptor T-cells, in identifying diseases and stages, responses to therapy, and whole-body bio-distribution in real-time, which leads to improvement in cancer patient management. In contrast, the long half-life of intact antibodies hampers their use as imaging agents due to the several days required for blood and background clearance in order to achieve a good signal-to-noise ratio. These emerging methods in PET may improve patient selection and target delineation and, ultimately, may become a useful tool for adaptive radiation planning as we collectively strive toward personalized medicine in radiation oncology [89,92].

### 5.5. PET-CT

The most widely available and widest molecular imaging modality used in oncology is PET-CT due to its non-invasive nature and high accuracy in its application and management in oncology. PET-CT is a quantitative technique that provides information about morphologically relevant, physiologic, and pathologic processes at the molecular level, as well as biodistribution, dosimetry, the limiting or critical organ or tissue, and the maximum tolerated dose (MTD). It could detect and quantify abnormal molecular activity throughout the body and have high accuracy in differentiating malignant tumors from benign ones. It can also be used to evaluate the response rates of chemotherapy to allow easy management and early detection of tumor recurrences. This is useful in order to identify non-responders as soon as possible and to modify treatment. Furthermore, radiation planning with a PET-CT scan can be more beneficial by modifying the radiation dose for patients with situs dose deposition in the tumor. It has the ability to determine the more active and metabolic areas within the tumor to direct more aggressive radiation to reduce the chance of converting to more aggression, which fulfils the potential of personalized medicine. Moreover, there are also dynamic PET/CT scans, which are a new technology of PET/CT scan that allows new opportunities for personalized nuclear medicine by providing better image quality in a short scan time that can be used to optimize administered radioactivity and for pediatric patients and sick patients who cannot remain still for long periods [2,48,59,75,95].

PET-CT scans have high sensitivity and specificity, allowing them to use radiopharmaceutical tracers such as F-18 fluorocholine, Ga-68, and C-11 methionine to measure cellular characterization and biological processes in a tumor at the molecular and cellular level. The ability to quantify the disease at a molecular level, tumor hypoxia, and bone metastases may help assess the global inhibitory effect of such multi-targeted therapeutic approaches. Notwithstanding, there is a lack of personalized radiotracers in PET-CT radiotracers, which presents a major limitation to the molecular imaging role in personalized medicine [11,48,72,95,96,97].

## 6. Conclusions and Future Direction

In conclusion, personalized medicine aims to enhance diagnostic precision and reduce therapeutic failures. Molecular imaging, which has emerged as a dynamic and exciting field of study, plays a great role in personalized medicine Figure 1 and Figure 2. Although many aspects of molecular imaging are still in their infancy, the long-term goal is for medical professionals to be able to use these techniques to make better diagnoses, make better treatment choices, and predict patient outcomes. It is anticipated that molecular imaging methods will experience even greater technological advancements in the next decade, which will ultimately impact personalized medicine in a significant way and become a new eye for medicine.

## Figures and Tables

**Figure 1 jpm-13-00369-f001:**
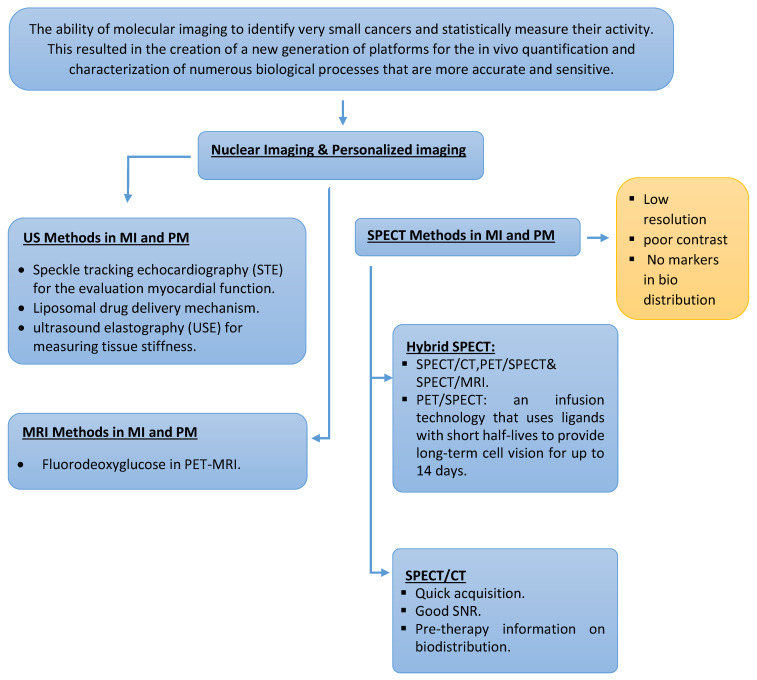
An improved knowledge of how a person’s particular molecular and genetic profile renders them prone to various diseases has resulted from scientific developments in customized medicine. The assessment of disease heterogeneity and progression planning, treatment, molecular features, and long-term follow-up are all common uses for molecular imaging techniques. The following illustration presents the key findings of this review.

**Figure 2 jpm-13-00369-f002:**
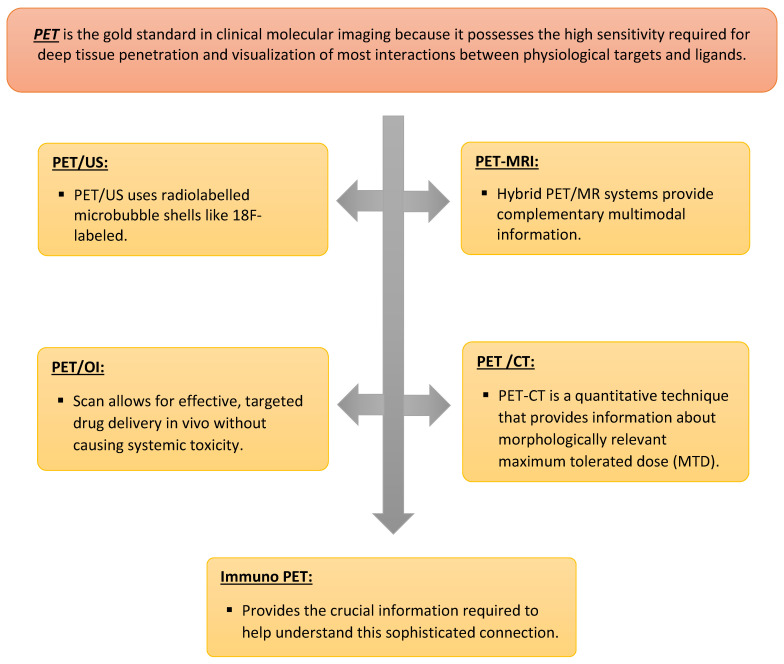
Positron Emission Tomography (PET).

**Table 1 jpm-13-00369-t001:** Common US microbubble contrast agents.

Trademark Name	Shell Material	Gas	Diameter in Volume (μm)	Microbubbles Volume Concentration (Bubbles/mL)	Recommended Dose (μm)	Application (Example)	Side Effects/Contraindications	References
Albunex	Sonicated serum albumin	Air	4.0	437 million/mL	0.033 to 0.5 mL/kg	Transpulmonary imaging, myocardial contrast echocardiography	Significantly increase thrombolysis with thrombolytics	[32,33,34]
Optison	Cross-linked serum albumin	Octafluropropane/Perflutren/Perfluoropentane	7.11 ± 0.24	0.078 ± 0.017 (×10^8^ mb/mL)	0.5 mL	Left ventricle opacification, endocardial	Hypersensitivity to perflutren, blood, blood products or albumin	[32,33,35,36,37]
Definity/Luminity	Phospholipids/DPPA/DPPC/MPEG5000 DPPE	Perfluoropropane/Octafluropropane/Perflutren	8.19 ± 0.77	0.143 ± 0.042 (×10^8^ mb/mL)	10 Ul kg^−1^	Echocardiography, liver and kidney imaging	Hypersensitivity to perflutren	[32,35,37,38]
Sonazoid	Hydrogenated egg yolk phosphatidyl serine (HEPS)/Phospholipid	Perfluoropropane	2.6 ± 0.1	1.27 × 10^9^ ppml	0.015 to 0.2 mL/kg	Myocardial perfusion, liver imaging, focal breast lesion	Diarrhea, albuminuria and neutropenia/Iodine Allergy and renal dysfunction	[32,33,38,39,40,41]
Lumason/SonoVue	Phospholipid/DPSC, DPPG-Na, palmitic acid	Sulphur hexafluoride	8.01 ± 0.85	0.022 ± 0.006(×10^8^ mb/mL)	4 × 10^7^ (bubbles/kg)	Left ventricle opacification, microvascular enhancement (liver, and breast lesion detection)	Hypersensitivity to sulphur hexafluoride or any inactive ingredient	[28,32,37,38]
Perfluorocarbon-exposed sonicated dextrose albumin (PESDA)	Dextrose albumin	Perfluorobutane	-	1.05 × 10^9^ mb/mL	2.5–10 (μL kg^−1^)	Carotid artery restenosis, Carotid intimal hyperplasia, liver, pulmonary	Thrombolysis	[35,42,43,44]

**Table 2 jpm-13-00369-t002:** Presents the SPECT radiotracers and their clinical applications.

Radiopharmaceutical	Modality	Clinical Applications	References
201TI chloride	SPECT	Brain tumors	[67]
99mTc-tetrofosmin	SPECT	Brain tumors	[68]
111In-DTPA-octreotide46	SPECT	Brain tumors, cerebrospinal fluid kinetics	[67]
99mTc-sulfur colloid	SPECT/CT	Splenosis, sentinel lymph node metastasis/biopsy	[52,67]
123I-iodine	SPECT/CT and SPECT	Neuroendocrine tumors	[52,67]
99mTc-sestamibi	SPECT/CT and SPECT	Breast cancer, lymph node metastasis	[52,67]
99mTc-diphosphonates	SPECT/CT and SPECT	Bone detection	[67,68]
99mTc-red blood cells	SPECT/CT and SPECT	Gastrointestinal bleeding and associated disorders: splenosis	[52,67]
99mTc MAA	SPECT/CT	Liver and lung pulmonary perfusion	[68]
99m Tc-N4-NIM	SPECT	Hypoxic	[47]
125 I-IPOS	SPECT	Hypoxic	[47]
99mTc-lablled	SPECT	Cardiac	[69]
111 In-oxyquinoline	SPECT	Stem cells visualize binding sites in receptor-expressing neuroendocrine tumors	[17,69]
I-131	SPECT/CT	Thyroid	[70]

**Table 3 jpm-13-00369-t003:** PET radiotracers and their role in some diseases and cancers.

Diseases/Disorders	Image Technique	Radiotracers	Applications	References
Pencentric Cancer	PET	[^18^F]-fluorodeoxyglucose (FDG)	Diagnosis, post-therapy monitoring	[7]
Prostate Cancer	PET, choline PET/CT, PET/MRI	[^18^F]-fluorodeoxyglucose [^18^F]-FDG [^11^C]-acetate 3′-Deoxy-3′-[^18^F]-Fluorothymidine (FLT) [^18^F]-2′-Fluoro-5-Methyl-1-β-D-Arabinofuranosyluracil (FMAU)	First-line diagnostic and staging procedure, guide biopsies and for planning of focal therapy, lymph node and bone metastases, evaluate therapy	[11,21,50,59,76,80,81]
Hematoma	PET, PET/CT	2-Deoxy-[^18^F]-fluoro-D: -glucose (FDG)	Detect the presence of hematoma associated with a malignant lesion, identify the hematoma that mimics a malignant tumor	[82,83,84]
Hypoxia	PET	[^18^F]-fluoromisonidazole (FMISO) [^18^F]fluoroazomycin-arabinofuranoside [^60/64^Cu]-copper(II)-diacetyl-bis(N4-methy- lthiosemicarbazone (ATSM)	Quantify chronic tissue hypoxia, calculate tumor HV and the maximum level of hypoxia, useful in radiotherapy metabolic planning, guiding the use of chemotherapeutic drugs	[58,59,83]
Glioma	PET	[^18^F]-fluorodeoxyglucose [^18^F]-FDG [^11^C]-methionine (MET) Fluorodeoxyglucose (FDG) Amnio Acid (AA)	Tumor recurrence detection and radiation necrosis, therapy monitoring, safe resection of glioma, identifying areas of infiltrating glioma, help optimize image-guided biopsy, radiotherapy planning	[21,76,77]
Cardiac	PET, PET/MRI, PET/CT	[^18^F]-fluorodeoxyglucose [^18^F]-FDG [^18^F]-sodium fluoride [^18^F]-NaF	Assess cardiac diseases: ischaemia detection and quantification, coronary calcification, and myocardial inflammation. Atherosclerosis detects endocarditis, infection of cardiac devices, and metastatic measures of inflammation in the vessel wall and myocardium injury, and monitor the therapeutic effect	[23,81,85,86]
Hodgkin’s Lymphoma	PET, PET/CT or PET/MR	[^18^F]-fluorodeoxyglucose [^18^F]-FDG	Staging and recurrence detection, evaluating the treatment response, assessment of a variety of types of lymphomas, sarcomas, and blastomas	[9,12,59]
Bone Cancer (Sarcoma)	PET, PET/CT	[^18^F]-fluorodeoxyglucose [^18^F]-FDG	Detect bone/bone marrow metastases, predict the therapy response	[75]
Fever of Unknown Region (FUO)	PET/CT	[^18^F]-fluorodeoxyglucose [^18^F]-FDG	Investigation and management in children with FUO guide the therapy drugs	[77]
RENAL MASS	PET, PET/CT	[^124^I]-girentuximab	Utilized for renal mass characterization, identification of ccRCC	[12]
Lung Cancer (NSCLC)	PET, PET/CT	[^18^F]-fluorodeoxyglucose [^18^F]-FDG	checkpoint blockade immuno- therapy, predict immunotherapy toxicity, mutational status, and metastases, guide decisions during therapy	[87]
BREAST CANCER	PET, PET/CT	fluorodeoxyglucose (FDG) 3′-Deoxy-3′-[^18^F]-Fluorothymidine (18FLT)	Staging, monitoring, and prediction of response to therapy agents	[80,88]
Alzheimer’s Disease (AD)	PET	[^18^F]-fluorodeoxyglucose [^18^F]-FDG N-[(2-[^11^C]-methoxyphenyl)methyl]-N-(6-phenoxypyridin-3-yl)acetamide-[^11^C]-PBR-28 [N,N-diethyl-2-(2-(4-(2[^18^F]fluoroethoxy)phenyl)5,7dimethylpyrazolo[1,5a]pyrimidin-3- yl)acetamide] [^18^F]-DPA-714	Early detection and treatment monitoring to expand our knowledge about the AD of different phenotypically	[83,84]

## Data Availability

Not applicable.

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
