# Peer review of "The Role of Molecular Imaging in Personalized Medicine"

_jpm, 2023, doi:10.3390/jpm13020369_

Round 1
Reviewer 1 Report
Synopsis:
The authors present a survey of on the role of molecular imaging in personalized medicine. This technology provides a unique window into the molecular changes that occur in disease, and can be used to monitor disease progression, guide therapy, and evaluate treatment response.
The language and writing quality of the manuscript is fair. However, the organization of the article can be improved with a figure summarizing different modalities that the authors survey is usually present in review articles and is currently missing in the manuscript.
The manuscript can be improved by considering the following comments:
General Comments:
1. Improve literature search: An important application of molecular imaging in personalized medicine is the ability to monitor treatment response in real-time. For instance, molecular imaging can be used to monitor the delivery and uptake of therapeutic agents, such as drugs and radiopharmaceuticals, in individual patients, and to evaluate treatment response in real-time. This information can be used to optimize therapy, adjust dosing, or switch to alternative treatments if necessary.
a. Focused Ultrasound for Drug Delivery:
https://www.fusfoundation.org/the-technology/mechanisms-of-action/drug-delivery-vehicles/
b. Ultrasound Elastography as a tool to map drug delivery: Clin Cancer Res. 2019 Apr 1;25(7):2136-2143. doi: 10.1158/1078-0432.CCR-18-2684. Epub 2018 Oct 23.
c. Ultrasound as a tool to monitor multiscale mechanical changes while tissue development:
S. Khan, S. Hollenbach, S. Goswami, F. Feng and S. A. McAleavey, "Placental Elastography reveal Viscoelastic Signatures ex-vivo with Single-Track Location Maximum a Posteriori Probability Spectroscopy," 2022 IEEE International Ultrasonics Symposium (IUS), Venice, Italy, 2022, pp. 1-4, doi: 10.1109/IUS54386.2022.9958924.
And many more.
2. A well-designed figure can help the reader quickly understand the key findings and comparisons of the reviewed studies.
Summarily, the authors report the use of Imaging in Personalized Medicine since molecular imaging is a powerful tool that is increasingly being used to advance personalized medicine by enabling the visualization of molecular and cellular processes in the body. This technology can transform how we diagnose, monitor, and treat diseases, ultimately improving patient outcomes. The article will benefit from a minor revision based on the above points.

Author Response
We agree, we have accordingly added it.

Reviewer 2 Report
1. Please add the references to the sentences “PET-US uses radiolabelled microbubble shells like 18F-labeled, albumin-shelled, and VEGFR2-targeted, which have a short half-life and several micrometres in size” (lines 314-315)
2. I think it’s a great review of molecular imaging in personalized medicine. I have no other comments. Congratulations.
Author Response
We agree, we have accordingly added it.
